# Development of the Menu Assessment Scoring Tool (MAST) to Assess the Nutritional Quality of Food Service Menus

**DOI:** 10.3390/ijerph20053998

**Published:** 2023-02-23

**Authors:** Claire Elizabeth Pulker, Leisha Michelle Aberle, Lucy Meredith Butcher, Clare Whitton, Kristy Karying Law, Amy Louise Large, Christina Mary Pollard, Georgina S. A. Trapp

**Affiliations:** 1East Metropolitan Health Service, Kirkman House, Perth, WA 6000, Australia; 2School of Population Health, Curtin University, Perth, WA 6845, Australia; 3School of Medical and Health Sciences, Edith Cowan University, Joondalup, WA 6027, Australia; 4The George Institute for Global Health, Sydney, NSW 2042, Australia; 5Enable Institute, Curtin University, Perth, WA 6845, Australia; 6Telethon Kids Institute, Nedlands, WA 6009, Australia; 7School of Population and Global Health, The University of Western Australia, Crawley, WA 6009, Australia

**Keywords:** obesity, food environment, food service, menu, policy, healthy eating, public health nutrition

## Abstract

Preventing the rise in obesity is a global public health priority. Neighbourhood environments can help or undermine people’s efforts to manage their weight, depending on availability of nutritious and nutrient-poor ‘discretionary’ foods. The proportion of household food budgets spent on eating outside the home is increasing. To inform nutrition policy at a local level, an objective assessment of the nutritional quality of foods and beverages on food service menus that is context-specific is needed. This study describes the development and piloting of the Menu Assessment Scoring Tool (MAST), used to assess the nutritional quality of food service menus in Australia. The MAST is a desk-based tool designed to objectively assess availability of nutrient-poor and absence of nutritious food and beverages on food service menus. A risk assessment approach was applied, using the best available evidence in an iterative way. MAST scores for 30 food service outlets in one Local Government Authority in Perth, Western Australia highlight opportunities for improvements. MAST is the first tool of its kind in Australia to assess the nutritional quality of food service menus. It was practical and feasible to use by public health nutritionists/dietitians and can be adapted to suit other settings or countries.

## 1. Introduction

Obesity is a common, complex, chronic, relapsing, diet-related condition [1]. Global prevalence continues to increase, with two billion people classified as obese in 2015 [2]. In Australia, two-thirds of adults were classified as overweight or obese in 2017–2018 [3]. Over one hundred different factors can contribute to weight gain and obesity, including biological, psychological, environmental, and social [4]. People who are classified as obese have an increased risk of chronic diseases including cardiovascular disease, type 2 diabetes, some cancers [5], as well as mental ill-health and eating disorders [6]. Preventing the rise in prevalence of excess body weight is a global public health priority [1,7]. In addition, supporting people who are classified as overweight or obese to reduce weight by 5–10 percent and maintain the weight loss can lead to significant risk reduction [5].

Public health interventions designed to reduce and prevent obesity include food and nutrition policies to improve population diets [8], as healthy dietary patterns are essential for achieving and managing a healthy weight [9]. Australian government policy addresses obesity via three dietary guidelines: Guideline 1: Achieve and maintain a healthy weight; Guideline 2: Enjoy a wide variety of nutritious foods; and Guideline 3: Limit intake of foods containing saturated fat, added salt, added sugars and alcohol [9]. In Australia, less than five percent of the population eat the types and amounts of foods recommended in the Australian Guide to Healthy Eating [9] to maintain a healthy body weight. Specifically, the most recent Australian Health Survey found that only four percent of adults ate the minimum recommended amount of vegetables and 35 percent of total dietary energy intake was from energy-dense-nutrient-poor or ‘discretionary’ foods (referred to as ‘nutrient-poor’ hereon in) [10].

Eating behaviour is influenced by multi-level factors, including individual (e.g., food preferences), social (e.g., social and cultural norms), environmental (e.g., neighbourhood food environments), and macro-level (e.g., government policies and programmes) [11]. A range of policy interventions to improve eating behaviours are needed, including those which aim to improve neighbourhood food environments [12].

### 1.1. Neighbourhood Food Environments

The food environment is multifactorial, including food access, provision, and price, as well as how food is labelled and marketed [13]. The term ’neighbourhood food environment’ is used to describe a mixture of food retail and food service outlets and is not limited to a residential neighbourhood [14]. Neighbourhood food environments can influence food purchases and consumption by the types of food outlets present and their location (the community food environment); and by the choice of foods available, their price, placement and promotion (the consumer food environment) [13]. They can help or undermine people’s efforts to manage their weight, depending on availability of nutritious and nutrient-poor foods [15]. There is a growing body of literature that describes Australian community [16,17] and consumer food environments [18]. However, little is known about Australian food service outlets, apart from cost and presence of nutrition information in selected fast food chains [19,20], and the nutritional quality of children’s menus from non-chain restaurants and cafes [21].

Food service outlets (e.g., cafes, coffee shops, restaurants, fast food or quick service, takeaway, pubs, hotels, and taverns) are important considerations for nutrition policy because they contribute an increasing proportion of total dietary intake in many countries, as more people are ‘eating out’ or ‘ordering in’ ready-to-eat meals or snacks. Australians purchased an average of 2.73 meals from outside of the home each week in 2018 [22]. A third of Australian households’ food expenditure was on restaurant meals and fast foods in 2015–2016 [23]. Similarly, in the United Kingdom (UK) household expenditure on food and drinks eaten outside the home comprised 32 percent of the total spent on food in 2019–2020 [24]. In the United States (US), households spent over half of their food budget on prepared foods from outside of the home in 2021 [25]. Meals purchased in restaurants or fast food outlets contributed a substantial proportion of US dietary energy intake but had poor nutritional quality [26]. Ecological studies indicate that growth in availability of fast food outlets may be associated with the increase in obesity, but this has not yet been established at a local or individual level [27].

### 1.2. Food Service Outlet Assessment Tools

Most of the existing tools to assess the impact of food service outlets on food environments and dietary health [28] require on-site audits. Food and nutrition policy interventions requiring on-site assessment of food outlets are time consuming, expensive, and unfeasible in Australia due to the widely dispersed urban areas [29,30,31]. To be practical and feasible for use on a large scale, a new tool to assess the impact of food service outlets on population diets without the need to conduct site visits is needed.

The Portuguese Kids’ Menu Healthy Score (KIMEHS) was developed as a desk-based tool and provides an easy and fast way to assess children’s menus for consistency with the Mediterranean diet [31]. KIMEHS penalises the availability of unhealthy items and assigns reward points for the availability of healthy items, providing clear signals for improvement to food outlet operators [31]. While some dietary advice is universal (e.g., to consume fruit and vegetables; and limit salt, sugar and fat) there are some culturally specific differences regarding the types and amounts of foods recommended between countries [32]. The Australian Guide to Healthy Eating was designed to inform common dietary patterns reflective of the country’s multi-cultural heritage, and recommendations are not specific to the Mediterranean dietary pattern [9]. Menu nutritional quality is an essential component of an objective tool to assess the impact of food service outlets on population diet, and can inform national and local policy interventions [33]. An objective menu assessment tool that is tailored for the Australian context is therefore needed.

### 1.3. East Metropolitan Health Service Assessment of Food Service Outlets

The East Metropolitan Health Service (EMHS), based in metropolitan Perth, is one of four publicly-funded area health services within Western Australian (WA). EMHS provides tertiary, secondary, specialist, and community and population health services that aim to maintain and improve the health of one-third of the WA population [34]. The EMHS Obesity Prevention Strategy uses evidence-based approaches to address overweight and obesity and its determinants via 28 priority actions [35]. It includes an action to “classify the dietary risk of food outlets, identify issues of concern, and inform and influence local public health responses” ([35], p. 16).

As part of the implementation of the EMHS Obesity Prevention Strategy, the Food Outlet Dietary Risk assessment tool (FODR) was developed to objectively assess and score the dietary risk of all consumer-facing food outlets present in EMHS neighbourhood food environments. It assigns scores for six public health nutrition attributes: availability of nutrient-poor foods; availability of nutritious foods; acceptability and appeal; accessibility; type of business operation; and complex food outlet considerations [36]. A risk rating similar to that used for food safety is assigned to each food outlet, from low to very high. Food outlet data for FODR can be obtained during routine environmental health officer site visits, or from desk-based research using the best available evidence. When pilot testing the FODR tool, the research team identified that a consistent method to objectively assess the nutritional quality of food service menus, specific to the Australian context, was needed.

This study aims to describe the development and piloting of the Menu Assessment Scoring Tool (MAST), to assess the nutritional quality of food service menus in Australia. MAST was designed using a risk assessment approach [36], to be used by public health nutritionists and dietitians to conduct desk-based audits of food service outlets. Risk assessment principles applied in the development of MAST were similar to those used in food safety risk assessment and included using the best available evidence in an iterative way and recognising the inherent uncertainty in risk assessment [37].

## 2. Materials and Methods

### 2.1. Development of the MAST

#### 2.1.1. Database of Classified Food Outlets

EMHS staff collected registered food business data from all 13 Local Government Authorities (LGAs) in 2018. A database of 6963 food businesses was constructed by the research team, and a classification framework applied (Appendix A). Food businesses classified as consumer-facing food retail (e.g., supermarkets, convenience stores) and food service outlets (e.g., cafes, restaurants) (*n* = 4136) were identified [36]. Other types of food service provision, including in schools, hospitals, or sports and recreation centres were classified as ‘institutional food’ to reflect the different ways in which people interact with them as well as different approaches to policy responses (e.g., mandatory nutrition criteria for school meals), and not included in this study.

#### 2.1.2. Principles for Assessment of Food Service Menus

MAST was developed to be fit-for-purpose in the Australian context, applying the following principles during its development: (1) menu items were classified as either nutritious (i.e., the types of foods recommended in the Australian Guide to Healthy Eating [9]) or nutrient-poor, consistent with national guidelines; (2) a risk assessment approach was adopted using the best available evidence, with menu items assumed to be nutrient-poor unless available information demonstrated otherwise; (3) the desk-based tool needed to be quick and easy to use; (4) face validity of menu scores assigned by the research team of qualified public health nutritionists and dietitians was required; and (5) areas for improvement should be signalled to food outlet operators. MAST is used to assess availability of nutritious and nutrient-poor food and beverages (referred to as food hereon in) on food service menus.

#### 2.1.3. MAST Categorisation of Menu Items

Details provided in the Australian Guide to Healthy Eating [9] and the accompanying Eat for Health Educator’s Guide [37] formed the basis of menu item classification as ‘nutritious’ or ‘nutrient-poor’ for MAST.

The Australian Guide to Health Eating recommends enjoying a wide variety of nutritious foods from five food groups every day, including: grain (cereal) foods; vegetables and legumes/beans; fruit; milk, yoghurt, cheese and alternatives; and lean meats and poultry, fish, eggs and alternatives. Plenty of water is recommended, and small amounts of healthy fats and oils can be used when preparing meals [9]. Discretionary, or energy-dense-nutrient-poor foods, should only be eaten sometimes and small amounts. Examples include sweetened drinks, processed meats, sausages, pies and pastries, commercially fried foods including chips, biscuits, cakes, puddings, and alcoholic beverages [37]. The Eat for Health Educator Guide recommends that the main ingredients of mixed foods or meals are identified and classified as nutritious five food group foods or energy-dense-nutrient-poor discretionary items for dietary analysis [37]. However, this approach does not categorise whole menu items as either nutritious or nutrient-poor and is problematic when only short meal descriptions are provided on food service menus. Nutritious five food group foods and water are referred to as ‘nutritious’ foods and discretionary energy-dense nutrient-poor foods are referred to as ‘nutrient-poor’ foods hereon in.

Lack of consistency in classifying foods as ‘discretionary’ has been identified among Australian health professionals, food industry, policy makers, and consumers [38]. Therefore, to achieve consistency among the research team and support wider translation of MAST, detailed definitions of food groups were developed. They were adapted from the Eat for Health Educator Guide and the 38 food groups used by Wang and colleagues to assess menu items available from an online delivery platform in Sydney in 2020 [38]. During its development, six public health nutritionists and dietitians and two dietetics students used MAST to assess menus from food service outlets present in Perth, WA. Development of food groups and definitions was done in a collaborative and iterative way to reflect the challenges of assessing a wide variety of food service menus, and to resolve any discrepancies in classification arising among the research team.

#### 2.1.4. MAST Scoring System

The MAST scoring system was adapted from the work of Tavares et al. (2021), which recognised that food service outlets supportive of healthy eating make nutritious choices available and limit nutrient-poor choices [39]. The Healthiness Indicator, developed in Brazil to characterise food outlets selling food for immediate consumption, assigns a reward point for each of the nutritious food groups sold, and for each of the nutrient-poor food groups not sold [39].

Using the risk assessment approach, MAST assigns a penalty point for the availability of each nutrient-poor food category, and for the absence of each corresponding nutritious food category. When one meal meets the definition of each nutrient-poor food category, one penalty point is allocated. When there are no meals available that meet the definition of each corresponding nutritious food category, one penalty point is allocated. There is no change to the penalty points allocated when more than one meal meets the definition of each nutrient-poor or corresponding nutritious food category. The scores assigned by MAST range from 0% to 100%. The best-case scenario produces a score of 0% which indicates there are no nutrient-poor food groups present on the menu, only nutritious food groups. The worst-case scenario products a score of 100% which indicates there are nutrient-poor food groups present on the menu, but no nutritious food groups (Figure 1).

Categorical risk (e.g., low, medium, high) is not assigned by the MAST score for food service menus. This is because the MAST score contributes two of the six components of the Australian FODR tool (i.e., availability of nutrient-poor food; availability of nutritious food) which was developed to assess and score the dietary risk of all consumer-facing food outlets present in EMHS neighbourhood food environments.

**Figure 1 ijerph-20-03998-f001:**
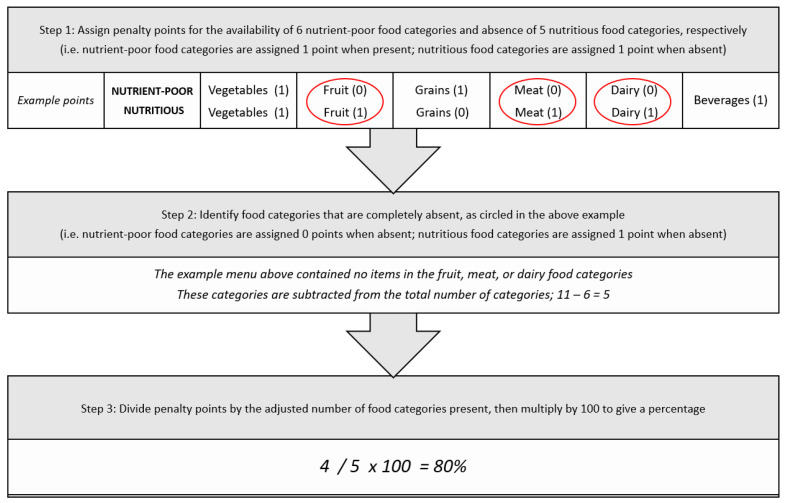
Schematic of the Menu Assessment Scoring Tool (MAST) scoring system.

#### 2.1.5. Sourcing Online Menus

Some food service outlets provided menus in multiple online locations, including one or more online delivery platforms (e.g., Deliveroo^®^ (Balaclava, VIC, Australia), UberEats^®^ (Port Melbourne, VIC, Australia)), company websites, company social media platforms (e.g., Facebook^®^ (Sydney, NSW, Australia)), company mobile phone apps, or user-generated review websites (e.g., TripAdvisor^®^ (Sydney, NSW, Australia)). To determine whether the menus available at food service outlets differed across online locations, a sample of five major chain food service outlets which sold a range of foods were identified for comparison. Chains were a focus for comparison because they have the most influence over neighbourhood food environments due to contributing a high proportion of food service sales, an approach recommended by the International Network for Food and Obesity/non-communicable diseases Research, Monitoring and Action Support (INFORMAS) [40]. The five chains sold burger-based products, chicken-based products, pizza, sandwiches, and Mexican-based products. The largest online delivery platforms were identified as UberEats^®^, Meunlog^®^ (Sydney, NSW, Australia) Deliveroo^®^, and Doordash^®^ (Melbourne, VIC, Australia) (all apps were iOS version 16.1) and used in the analysis. In total, 26 menus were collected in October 2022 for comparison as follows: 5 from UberEats^®^, 4 from Menulog^®^, 4 from Deliveroo^®^, 4 from Doordash^®^, 4 from company websites, and 5 from company mobile phone apps. A standard location in the EMHS geographic catchment was used across all online delivery platforms, websites, and apps. All five chains received identical MAST scores for each of the menus available online. Based on this comparison, the research team agreed that using one online delivery platform as the default for obtaining food service menus would provide robust MAST scores.

### 2.2. Pilot Testing MAST

#### 2.2.1. Identify Food Service Outlets Present

One LGA agreed to provide an updated database of registered food businesses in 2021, which formed the setting for pilot testing of MAST. The data were cleaned and classified and included 24 food retail and 38 food service outlets.

#### 2.2.2. Source Menus Online

Online searches were conducted to locate menus for each of the 38 food service outlets. Online delivery platforms were used as the primary source of menus, supplemented by food outlet websites and social media. Two food service outlets appeared to have closed, no recent menus could be located for a further six, and 30 menus were obtained.

#### 2.2.3. Pilot Testing MAST Categorisation of Menu Items

The menus were imported into NVivo (QSR International Pty Ltd., Release 1.6.1, Doncaster, Victoria, Australia) and saved in a predetermined filing system. A MAST coding framework was developed and applied to each menu, which identified availability of six nutrient-poor food categories (which include 15 nutrient-poor food groups), and five nutritious food categories (which include 12 nutritious food groups). NVivo served as a storage platform for web captures of menus and visual records of the assessment decision making process.

For menu items that could be tailored by the customer, e.g., by adding a topping or changing the side dish, the default option was used. For menus with meal promotional packages or ‘meal deals’ the item was analysed together and not split into components. Menu items for children (e.g., kid’s menus) were included if listed on the main menu. MAST assigns a penalty point for each nutrient-poor food category available, so only one item per food category was coded in NVivo to indicate presence. MAST assigns a penalty point for each corresponding nutritious food category that is absent, so only one item per food category was coded in NVivo if available.

Availability of the food categories was then entered into REDCap^®^ (REDCap consortium, Version 12.4.21, Nashville, Tennessee, United States of America). REDCap^®^ housed the MAST tool and functioned as the quantitative data capture application. The presence or absence of each food group was entered into REDCap^®^ for each menu, using a MAST survey to assign scores as previously described, and the software then calculated the final score (Figure 2). Use of software and applications ensured a rigorous process was used by the research team to score the menus, which could be easily adapted during the iterative process undertaken.

#### 2.2.4. Pilot Testing the MAST Scoring System

The MAST scoring system was pilot tested with diverse food service menus during its development. Examples included: fine-dining restaurants requiring selection of a separate side dish (e.g., salad) to accompany the main dish (e.g., steak); cafes selling a wide range of teas and coffees but a small number of food offerings; fast food outlets with menu items that could be tailored with a choice of toppings or fillings; and specialty food service outlets such as ice cream and gelato shops. The MAST scores were reviewed to see what signals were given to food outlet operators for improvement.

Initial testing of the scoring system found that food outlets with nutrient-poor products from only one or two food categories (e.g., burgers, ice cream, fish and chips) would be advantaged over food outlets with more diverse menus. This was because they were allocated fewer penalty points due to the absence of multiple nutrient-poor food categories, despite selling few if any nutritious menu items. Based on this feedback, the final MAST scoring system was applied in three stages. Step one is to assign penalty points for the availability of each of six nutrient-poor food categories, and for the absence of each of five corresponding nutritious food categories, with the total used as the numerator for the MAST score (i.e., a maximum score of 11). Step two is to identify how many of the food categories received a penalty point, which is used as the denominator for the MAST score. In step three, the numerator is divided by the denominator and multiplied by 100 to give a percentage. This method produces more appropriate signals for improvement to a wide range of food service outlets.

**Figure 2 ijerph-20-03998-f002:**
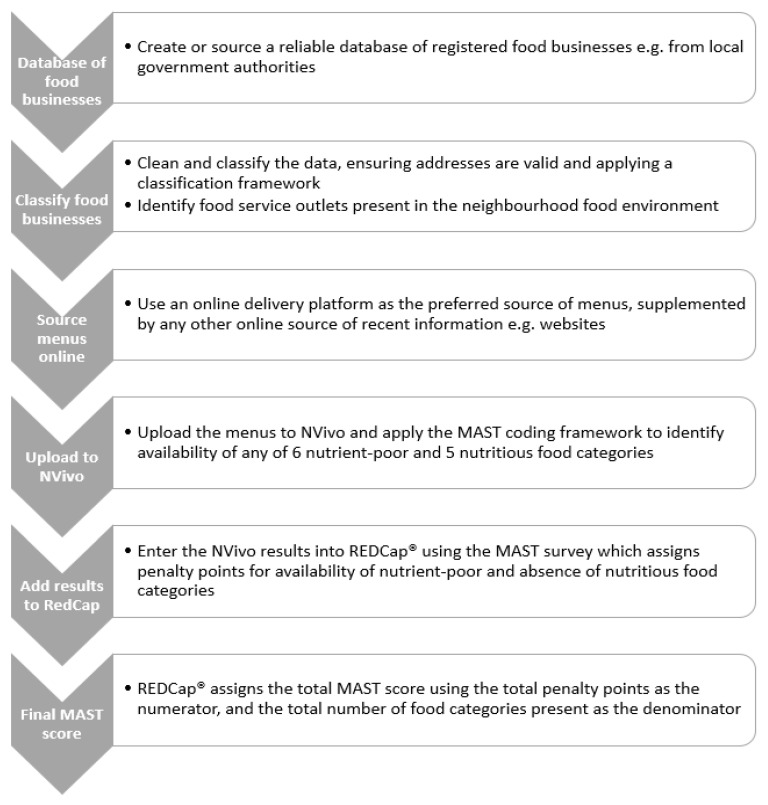
Menu Assessment Scoring Tool (MAST) process.

## 3. Results

### 3.1. The Final MAST

The final 27 food groups defined in MAST are mapped to six food categories of vegetables, fruits, grain (cereals), meat and alternatives, dairy and alternatives, and beverages and miscellaneous; and further defined as either nutritious (five food categories, 12 food groups) or nutrient-poor (6 food categories, 15 food groups) to reflect differences in ingredients and methods of preparation (Table 1).

Water was not included as a nutritious food group, because tap water is freely available in all Australian food outlets as a cultural norm. Alcoholic beverages were not included as a nutrient-poor food group, because the availability of alcohol is assessed separately in the overall FODR tool, via assessment of the type of liquor licence granted. It was therefore omitted from MAST to avoid duplication. If necessary, MAST may be modified for other settings to include water as a nutritious food group and alcoholic beverages as a nutrient-poor food group in the beverages and miscellaneous food category. For each food group, a definition and examples or exclusions are provided (Appendix A).

**Table 1 ijerph-20-03998-t001:** MAST food categories and nutrient-poor and nutritious food groups.

Food Category	Nutrient-Poor Food Groups	Nutritious Food Groups
**Vegetables**	**Vegetable-based mixed dish** **Fried potato (or similar)** Examples: Deep fried vegetable patties or croquettes, fried or oil-cooked potato products, salads with large amounts of creamy dressing	**Vegetable-based mixed dish** **Baked, steamed or roasted potatoes** Examples: baked, steamed or roasted potatoes, salads with modest amount of dressing, vegetable-based curries, stews, or casseroles, vegetable patties, stir fries with or without meat/alternatives, broths, blended and chunky soups
**Fruit**	**Fried fruit** **Juice and fruit-based smoothies** Examples: Fried fruit, fruit-based smoothies or juice containing added sugar or visible discretionary ingredients e.g., ice cream, cream, syrups/flavours, chocolate confectionary toppings, and any other sweeteners	**Fruit** **Juice and fruit-based smoothies** Examples: fresh fruit, canned/stewed/dried fruit, fruit or vegetable juices or smoothies with fruit as major component, with no added sugars
**Grain (cereals)**	**Cereal-based mixed dish** **Sweet or savoury baked goods/desserts** **(homemade or similar)** Examples: deep fried foods, meat pies, quiche, sausage rolls, pizza, pasta/lasagne with cream-based sauce, ramen, soy-sauce based soups, garlic bread, cookies, cakes, cake-type desserts, muffins, slices, sweet pies, scones, scrolls, brioche, pancakes/crepes with discretionary toppings, waffles, souffle, pastries	**Cereal-based mixed dish** **Breads and cereals** Examples: All breads and cereals, pasta, noodles, roti, bread rolls, flat breads, oatmeal/ porridge, crumpets, mixed meals with cereal products as the major ingredient and no discretionary ingredients including pasta, pizza, burgers, sandwiches/rolls/buns, sushi, wraps, rice, noodles, dumplings, steamed dim sum, burrito, taco
**Meat and alternatives**	**Meat and poultry, and/or meat and/or poultry based mixed dish** **Seafood and/or seafood based mixed dish** **Meat alternatives and/or meat alternatives-based mixed dish** Examples: Cured/salted/smoked/processed meats such as bacon, ham, salami, luncheon meats, sausages, dishes that are deep fried, contain coconut-based or cream-based sauce, beef stroganoff, chicken parmigiana, schnitzel, processed plant-based products mimicking meat	**Meat and poultry, and/or meat and/or poultry based mixed dish** **Seafood and/or seafood based mixed dish** **Meat alternatives and/or meat alternatives-based mixed dish** Examples: all raw, steamed and grilled seafood, lean meat, poultry, legumes, eggs, tofu, mixed dishes with meat/alternatives as the major ingredient and no discretionary ingredients including curry, stew, stir fry, casserole, meatballs, kebabs using cubes of meat, broths, blended and chunky soups, omelette, frittata
**Dairy and alternatives**	**Discretionary milk-based beverages** **Fried dairy-based foods** **Iced confectionery and dairy-based desserts** Examples: Milk-based drinks made with syrups, confectionery, ice cream, whipped cream, or sago pearls, dairy-based desserts that are deep fried, ice blocks, slushies, snow cones, jelly, frozen yoghurt, ice cream, gelato, sorbet, rice pudding, fromage frais, mousse, custard, iced drink desserts, panna cotta	**Flavoured milk/milk alternative based beverages** **Dairy and alternatives** Examples: Milk within dishes, milk on cereal, yoghurt, cheese and/or their alternatives, all fat levels of plain milk, rice milk, almond milk, macadamia milk, soy milk, drinkable yoghurts, milk-based smoothies, chocolate flavoured milk, iced/hot coffee or chocolate without ice cream/syrup/confectionery, latte, cappuccino, flat white, dirty chai
**Beverages and miscellaneous**	**Energy Drinks** **Sweetened and Rehydration Beverages** **Confectionery** Examples: energy drinks, soft drinks, cordial, non-dairy chain tea varieties, iced tea, mineral water with added sugar, lollies, chocolate, nougat, fruit leather, sesame snaps, peanut brittle, chocolate coated fruit/nuts/seeds, chocolate hazelnut spreads, chocolate sauces	n/a

n/a: Not relevant.

### 3.2. MAST Scores Assigned

MAST scores for 30 food service outlets present within the food environment of one LGA in the EMHS geographic catchment are summarised below (Table 2).

The mean MAST score across all 30 food service outlets was 71% (range 29–100%). Almost all food service outlets (24/30) had a MAST score of 56% or higher, indicating the menus were high risk and poor nutritional quality. The mean MAST scores for each type of food service outlet were: restaurants 56% (range 43–67%); cafes and coffee shops 56% (range 36–78%); pubs, hotels, and taverns 69% (range 57–78%); and fast food and takeaway 81% (29–100%). Only one food service outlet, which was classified as fast food or takeaway, had a MAST score below 30%, indicating few nutrient-poor food items were available on the menu. Five food service outlets, all classified as fast food or takeaway, had a MAST score of 100% indicating no nutritious menu items were available.

## 4. Discussion

MAST was developed as an objective tool for assessing the nutritional quality of food service menus in Australia, using a risk assessment approach. To ensure MAST was fit-for-purpose in the local context, a number of principles were identified as important during its development and pilot tested. They informed refinements to MAST, which were made using a collaborative and iterative process by the research team of qualified public health nutritionists and dietitians.

The first principle identified was to classify menu items as either nutritious or nutrient-poor, consistent with national guidelines, to attribute dietary risk. Food classification requires the skills and expertise of public health nutritionists or dietitians [36], however discrepancies in classification can still occur due to the absence of detailed definitions of mixed foods and meals [41]. Creating detailed definitions for MAST ensured an agreed, consistent approach was used, aligned to current government policy. This will support wider dissemination and translation in Australia. Refining MAST definitions to include examples of common dishes found on menus, and resolving discrepancies in classification between the research team, aimed to improve accuracy and ease of use. The definitions were aligned to the Australian Dietary Guidelines, published in 2013, which are currently under review. When revised guidelines are published at the end of 2025 [42], food groups and definitions included in MAST will need to be updated to reflect any changes. The MAST could also be adapted for use in other countries by aligning the definitions to national food-based dietary guidelines.

The second principle used in the development of MAST was to adopt a risk assessment approach, using the best available evidence [43]. Menu items were assumed to be nutrient-poor unless available information demonstrated otherwise. The widely used NEMS-R also adopted this approach to identifying healthy menu items [44]. For example, in the absence of nutrition information or a regulated claim (e.g., low-fat) on menus, the NEMS-R only assigned salads as healthy if low-fat or fat-free dressings were used and a maximum of two salad ingredients were over 50% fat [44]. Pilot testing of MAST identified some practical considerations, including sourcing of food service outlet menus online. Some smaller, non-chain food service outlets had minimal or no online presence which meant they were unable to be assessed using MAST. For future analysis, the number of missing menus will inform the approach used. When there are only a small number of missing menus, EMHS or LGA staff may visit the venues and request menus for assessment. When the number of missing menus is larger, the mean MAST score for each type of food service outlet will be assigned as the best available evidence.

The third principle that was essential for feasibility of applying MAST to EMHS neighbourhood food environments, was for the desk-based tool to be quick and easy to use. Using menus that were sourced online reduced the time needed to undertake site visits, which have been reported to take up to 40 min per outlet when using the NEMS-R [30]. Also, application of MAST does not require every menu item to be classified, as only one item from each food category is coded to demonstrate whether it is available or absent. The two steps which used NVivo software for coding and a REDCap^®^ survey for scoring added to the time taken for assessment but ensured consistency and transparency. Verification of coding between different EMHS staff could be undertaken, and changes made during the iterative development process. It was also envisaged that the documents saved during the process of deriving MAST scores could be used to provide tailored feedback to food service outlets seeking to improve menu nutritional quality in future. Therefore, the process adopted for MAST was considered time efficient and robust.

The fourth principle for developing MAST was for face validity in the scores assigned by the research team. Face validity was assessed by identifying whether MAST appeared to do the job it aimed to do, from the users’ perspectives [45]. MAST was developed by EMHS staff over a period of 18 months, involving contributions from six public health nutritionists and dietitians and two student dietitians. The collaborative and iterative process included regular meetings to discuss the scores assigned and resolve any discrepancies in classification of menu items. For example, due to the cultural status of meat in Australia some staff assumed any mixed dish that included meat would be coded to the meat and alternatives food category. However, for dishes where cereal or grain foods were a main ingredient including burgers and pizza, MAST assigns them to this food category. Changes to MAST definitions over the period of development led to common understanding and agreement among the research team. MAST was deemed to have achieved face validity by EMHS staff (i.e., the users) because the tool categorised menu items using definitions adapted from the Australian government’s Eat for Health Educator Guide [37]. It could be applied to objectively identify presence or absence of menu items that were consistent with government recommendations.

The fifth principle that was identified as important when developing MAST was to signal areas for improvement to food service outlet operators. The way the MAST score is calculated means that when one meal meets the definition of a nutritious food category penalty points are not allocated. In contrast, if numerous meals meet the definition of a nutrient-poor food category only one penalty point is allocated. Therefore, the strongest signal to food service outlet operators is to add nutritious meals to the menu rather than to reduce the number of nutrient-poor meals, although both changes are recommended. An Australian study which used an adapted NEMS-R to assess 28 rural food service outlets recommended urgent action to introduce and promote nutritious meals, as the outlets were dominated by nutrient-poor foods [46]. People engaged in weight management strategies reported avoiding food service outlets due to poor availability of nutritious food, and abundant availability of nutrient-poor options [15]. Supporting food service outlets to improve availability of nutritious food on menus should be a policy priority for regions aiming to address obesity, which is reinforced by MAST. Using MAST to screen all food service outlets present in a neighbourhood food environment could inform design of future interventions to support food service outlets to make positive changes.

Strengths of MAST include the ability to use the objective results to create tailored reports for food service operators willing to make changes to menus. The results can highlight where menus are supportive of healthy eating, and areas for improvement. A toolkit of suggestions can be constructed based on EMHS staff experience of supporting implementation of a mandated food and nutrition policy across health service sites [47]. A quality improvement approach could be used that identifies quick wins (e.g., adding steamed vegetables as a side dish) or easy wins (e.g., removing processed meat from an otherwise nutritious sandwich), which would require resourcing at EMHS or LGAs. MAST could also be adapted for use in different settings, including to encourage more nutritious food options at community food events organised by LGAs, such as farmer’s markets and food truck events. MAST can be used as a stand-alone tool to assess the nutritional quality of menus, or to contribute scores to a more comprehensive assessment of dietary risk such as the Australian FODR assessment tool.

The study also has some limitations. Food service outlets change over time, and menus also change depending on seasonality, ingredient cost and availability, and customer preferences. MAST assessment can be repeated over time but will not capture all of these ad hoc changes. MAST is used to assess nutritional quality of menus using only the information provided. Therefore, cooking methods which include adding significant amounts of fat, sugar and salt may not be identified. MAST does not include assessment of prices, promotions, or portion size. Sourcing menus from online food delivery platforms means there may be some differences in the offer available, compared to visiting the physical food service outlet. This is more likely to occur for smaller and independently operated food service outlets, who may have less capacity to ensure online information is updated regularly. The indicative MAST score for food service menus does not incorporate the comprehensive assessments used in other settings with mandated government policies, such as the Healthy Options WA Food and Nutrition Policy in hospitals [48]. MAST was developed as a rapid desk-based assessment of food service menus, suitable for use in large-scale surveillance of neighbourhood food environments. It can be used as a screening tool for risk, rather than a tool that provides a precise and detailed assessment of every menu item. As demonstrated in this study, MAST was able to detect variation between food outlets despite this limitation. It was developed to be used alongside the FODR tool which also assesses acceptability and appeal; accessibility; type of business operation; and complex food outlet considerations [36]. The risk-based approach, which assumed items were nutrient-poor unless information was available to demonstrate otherwise, may have resulted in over-estimation of risk in some food service outlets. However, given that presence of only one nutrient-poor item was required for a penalty point to be assigned, the chance of over-estimation of risk is unlikely.

## 5. Conclusions

The new MAST is the first tool of its kind in Australia to assess the nutritional quality of a wide range of food service menus. MAST was used to accurately and objectively assess availability of nutritious and nutrient-poor food on food service menus. MAST was practical and feasible to use by qualified public health nutritionists and dietitians and can be adapted to suit other settings or countries. Measuring availability of nutrient-poor food and absence of nutritious food contributes two of six components of the Australian FODR tool which assesses the dietary risk of consumer-facing food outlets. Pilot testing MAST with one LGA within Perth, WA identified the typically poor nutritional quality of menus. Findings indicate the challenges to eating healthily faced by people who are managing their weight. Assessing the nutritional quality of menus from neighbourhood food environments across the EMHS geographic catchment using MAST will inform local policy responses addressing obesity.

## Figures and Tables

**Table 2 ijerph-20-03998-t002:** MAST scores for food service outlets located in one LGA in the EMHS geographic catchment.

Food Service Outlet Number	Food Service Outlet Classification	MAST Numerator ^1^	MAST Denominator ^2^	MAST Score ^3^
1	Fast casual/quick service/takeaway	2	7	29%
2	Café/coffee shop	4	11	36%
3	Fast casual/quick service/takeaway	2	5	40%
4	Restaurant	3	7	43%
5	Café/coffee shop	5	11	45%
6	Fast casual/quick service/takeaway	5	11	45%
7	Fast casual/quick service/takeaway	5	9	56%
8	Café/coffee shop	4	7	57%
9	Pub/hotel/tavern	4	7	57%
10	Restaurant	4	7	57%
11	Restaurant	4	7	57%
12	Café/coffee shop	7	11	64%
13	Pub/tavern/bar/winery/distillery	7	11	64%
14	Restaurant	6	9	67%
15	Café/coffee shop	7	9	78%
16	Pub/hotel/tavern	7	9	78%
17	Pub/hotel/tavern	7	9	78%
18	Fast casual/quick service/takeaway	4	5	80%
19	Fast casual/quick service/takeaway	9	11	82%
20	Fast casual/quick service/takeaway	6	7	86%
21	Fast casual/quick service/takeaway	6	7	86%
22	Fast casual/quick service/takeaway	6	7	86%
23	Fast casual/quick service/takeaway	10	11	91%
24	Fast casual/quick service/takeaway	10	11	91%
25	Fast casual/quick service/takeaway	10	11	91%
26	Fast casual/quick service/takeaway	9	9	100%
27	Fast casual/quick service/takeaway	9	9	100%
28	Fast casual/quick service/takeaway	11	11	100%
29	Fast casual/quick service/takeaway	9	9	100%
30	Fast casual/quick service/takeaway	11	11	100%

Note: ^1^ = total penalty points for the availability of each of six nutrient-poor food categories, and for the absence of each of five corresponding nutritious food categories (i.e., a maximum score of 11); ^2^ = the total number of food categories present (i.e., maximum score of 11); ^3^ = the numerator divided by the denominator and multiplied by 100 to give a percentage.

## Data Availability

The Menu Assessment Scoring Tool (MAST) is provided as Appendix A.

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
