# Peer review of "Development of the Menu Assessment Scoring Tool (MAST) to Assess the Nutritional Quality of Food Service Menus"

_ijerph, 2023, doi:10.3390/ijerph20053998_

Round 1

Reviewer 1 Report

Please see below for comments aimed at improving the clarity and quality of the final manuscript.

1. Line 64 - update to "neighborhood".

2. Introduction - section 1.2 does not appear to help frame the research question/Aims and reads more like a thesis literature review of available methods currently. Ideally, the Introduction should contextualise your work to the wider readership of IJERPH. Consider summarising this content to align with introducing your research. If I understand correctly, the most pertinent point here is that previous methods of menu assessment have been developed elsewhere in the world but these are not usable in the Australian context due to differences in e.g.  overarching dietary/service guidelines?

3. Lines 138-140 - while this statement highlights how you will use the term "food" henceforth, it also appears to currently read as a conclusion and so is out of place here. I suggest removing this statement and adding the pertinent statement (i.e. that the term "food" is used to describe food and beverages from food service) to the Methods section.

4. Methods - all software (e.g. NVivo) and online systems (e.g. REDCap®) should include detail of the manufacturer/designer, along with the city/country of origin.

5. Methods/workflow - there is limited rationale to explain why the MAST process includes menu entry first into NVivo, then into REDcap. The functionality of these systems is not explained to the wider readership. From my perspective, I understand that NVivo is standardly used to process qualitative data while REDCap is a broader databasing tool. The content does not help me understand what is being carried out with both systems and why both are necessary at the moment. Please consider including additional rationale to make the process clearer to a wider readership. I also note that the "Add Results to RedCap" component of Figure 1 contains the word "nutient" which needs to be updated. "REDCap" should be used consistently throughout the manuscript as well - this is presented in a different format in Figure 1.

6. Lines 286-290 - is this statement correct? Presumably, the aspects of what might meet any given countries' guidelines of being defined within the "nutritious" or "nutrient-poor" food groups could be different. For example, food-based guidelines in one country might mean that e.g. "roast potatoes" are not considered a positive food choice

In Table 1, the option that perhaps sticks out is the flavoured milk/milk alternative category. Should this include [non-flavoured] "milk/milk alternatives" as well?

7. As a more minor point, my understanding is that AHEG refer to recommended food items as "nutritious" but other items as "discretionary" rather than "nutrient-poor". It may be helpful to align your terminology (in relation to the two categories in Table 1 and elsewhere in the manuscript) to align with these terms for clarity. There are a number of items in Table 1 that can be defined as "discretionary" but not necessarily "nutrient-poor".

8. Figure 2 - this would appear to be only necessary as a caption for Table 1 to add clarity. The image itself looks to be screen-captured. Please add relevant explanatory text to a caption for Table 1 or elsewhere to improve presentation quality.

9. Methods - the approach and rationale for calculating the MAST Numerator and Denominator should be clearly defined in the Methods section, rather than only summarised in a Results table (Table 2). This scoring system would appear to suggest that an "ideal" menu will have 50% of the options falling into the "nutritious" category. This would appear to allow comparison across different outlets/menus but this approach needs rationalised in much greater detail in terms of usability and relevance to AHEG or food service standards/recommendations. Presumably it is possible from this approach to score higher than 100% (and possibly even infinity if zero less positive items are included in the menu)?

10. Line 321 - "consistent with national dietary guidelines". See points 7 and 9 above. A score of 100% does not suggest the food service won't result in discretionary choices, only that these do not outweigh more prudent options.

11. Lines 331-333 - see point 6 above. This statement does not align with your previous assertion. I recommend removing the statement made in line 286-290, as the statement here appears to be more appropriate.

12. Lines 369-371 - it's unclear to me how face validity has been achieved here. How is the broad effectiveness of this approach being assessed? Has the approach helped to quantitatively separate menus that might be considered by practitioners as being qualitatively "better" or "worse" than each other.

Author Response

Thank you for taking the time to review our manuscript and provide constructive comments. The feedback has helped us to strengthen the manuscript by adding more detailed information, explaining the development, use, and pilot testing of the MAST. We have also clarified its limitations when compared to more detail-orientated menu audit tools used in other countries and settings.

Please find our responses to each comment in the table attached.

The updated manuscript highlights the revisions via tracked changes.

We hope that we have addressed all concerns and suggestions raised.

Reviewer 2 Report

Thank you for the opportunity to review the manuscript ‘Development of the Menu Assessment Scoring Tool (MAST)  To Assess the Nutritional Quality of Food Service Menus’ it is good to see a focus on food service in the effort to improve the health of food environments, as this has been a neglected area. The development of a tool that can measure the health of these environments remotely is also an advancement in the area.

The introduction provides a thorough overview of the background, though in the paragraph beginning line 59 it would be good to see a full acknowledgement of identified factors that influence food choice, e.g. advertising in non traditional media (social media advertising of food products and quick service restaurants), cost, access, and availability.

I wonder if the use of the term ‘neighborhood food environments’ is exactly correct? An individuals choices are influence by any food environment that they are choosing food in- not necessarily confined to their neighborhood, especially in such a geographically disperse country as Australia. Also, with menus being collected digitally it seems to be a moot point to use this term?

In regards to frequency of consuming food purchased out of home in Australia there is a more recent paper that you may wish to include (Consumption Frequency and Purchase Locations of Foods Prepared Outside the Home in Australia: 2018 International Food Policy Study. A. J. Cameron, L. H. Oostenbach, S. Dean, E. Robinson, C. M. White, L. Vanderlee, et al.).

Is there a definition of what is considered a ‘food service outlet’? Distilleries are included in this definition in Sup table 1, though is there an assumption that they must also serve food? Similarly, several of the businesses listed in the ‘Institutional Food’ category are traditionally considered to be food service outlets (e.g., hospitals) and those included in the ‘Accommodation and recreation services’ also often contain food service outlets (e.g., sports and recreation centres). Has the used framework been validated or applied elsewhere?

Sentence beginning line 98- reference?

Sentence beginning line 107- is there a specific aspect of the Portuguese tool that doesn’t correspond to an Australian setting? It would be good to clarify why this tool is not appropriate to measure the healthiness of food service environments in Australia.

What was the rationale for the decision ‘menu items assumed to be nutrient- poor unless available information demonstrated otherwise’?

Scoring system:

Please clarify if outlets that sell a narrow range of foods (e.g. a sushi shop) are penalised for not selling items from the nutritious food list?

Sentence beginning line 199: does this mean that two stores that sell 1. 1 nutrient poor vegetable item and 5 nutritious vegetable items; and 2. 10   nutrient poor vegetable items and 1 nutritious vegetable item receive the same penalty for that category?

Was the validity of the tool determined by comparing scores determined via online menu and from visiting the food service outlet in person?

Sourcing online menus- would the major food service chain outlets not have a more likely chance of having consistent menus across platforms due to their staff capacity allowing for updating these? Why wasn’t a random sample of retailers (including independent food service outlets) used?

By measuring only retailers that have online menus, is this not more of a measure of online neighborhood food environments than what is actually available?

The overall scoring system is difficult to understand in the current format, a schematic of how it should be applied would be easier to interpret.

What was the rationale for not having a nutritious equivalent of ‘sweet or savoury baked goods/deserts’ besides from ‘bread and cereals’?

Table 1- as currently presented, this table does not show any discernible difference between what is considered nutrient poor or nutritious for multiple examples. This table requires further information to be included in the final manuscript.

Line 376: this seems in opposition to a lot of the advice provided to retailers on how they can improve the health of their menus (by improving the healthiness of existing items by substituting unhealthy for healthier ingredients or using smaller amounts of unhealthy ingredients). Would the MAST tool be able to discern if these changes have been made?

Does MAST or the FODR process look at price or promotion? This needs to be addressed in the discussion.

Line 386: From the included information this seems to require the caveat that retailers must have online menus to provide this tailored advice. Has it been considered that it would be smaller retailers (in comparison to large chains) that may be willing to make changes to their menus, and that they may be the ones least likely to have the capacity to keep online menus up to date (if they do have online menus)?

Author Response

(The authors gave the same response as above.)

Round 2

Reviewer 1 Report

I'd like to thank the authors for responding to all reviewer comments efficiently and positively. The additions have greatly improved the clarity of presentation.